# Iron–Carbonate (Bi, Cu, Li) Composites with Antimicrobial Activity After Silver(I) Ion Adsorption

**DOI:** 10.3390/toxics13100825

**Published:** 2025-09-27

**Authors:** Alexandra Berbentea, Mihaela Ciopec, Adina Negrea, Petru Negrea, Nicoleta Sorina Nemeş, Bogdan Pascu, Paula Svera, Narcis Duţeanu, Cătălin Ianăşi, Orsina Verdes, Mariana Suba, Daniel Marius Duda-Seiman, Delia Muntean

**Affiliations:** 1Faculty of Chemical Engineering, Biotechnology and Environmental Protection, Politehnica University Timisoara, Vasile Parvan Blvd, No. 6, 300006 Timişoara, Romania; alexandra.berbentea@student.upt.ro (A.B.); adina.negrea@upt.ro (A.N.); petru.negrea@upt.ro (P.N.); narcis.duteanu@upt.ro (N.D.); 2ISIM-National Research and Development Institute for Welding and Material Testing, Mihai Viteazul Blvd., No. 30, 300222 Timisoara, Romania; 3Research Institute for Renewable Energies—ICER, Politehnica University Timisoara, Gavril Musicescu Street, No. 138, 300774 Timisoara, Romania; ioan.pascu@upt.ro; 4National Institute for Research and Development in Electrochemistry and Condensed Matter, Dr. A. P. Podeanu Street, No. 144, 300569 Timisoara, Romania; paulasvera@gmail.com; 5Coriolan Drăgulescu Institute of Chemistry, Mihai Viteazul Blvd, No. 24, 300223 Timisoara, Romania; ianasic@acad-icht.tm.edu.ro (C.I.); orsinaverdes@acad-icht.tm.edu.ro (O.V.); mariana_suba@acad-icht.tm.edu.ro (M.S.); 6Department of Cardiology, Victor Babes University of Medicine and Pharmacy Timisoara, Eftimie Murgu Square No. 2, 300041 Timisoara, Romania; daniel.duda-seiman@umft.ro; 7Multidisciplinary Research Centre on Antimicrobial Resistance, Department of Microbiology, “Victor Babes” University of Medicine and Pharmacy, Eftimie Murgu Square, No. 2, 300041 Timisoara, Romania; muntean.delia@umft.ro

**Keywords:** silver, recovery, adsorption, carbonates, bismuth, copper, lithium, iron, antimicrobial activity, antifungal agent

## Abstract

In the present study three composite materials based on iron in combination with bismuth, copper or lithium carbonates FeNO_3_@Li_2_CO_3_ (SFL), FeNO_3_@CuCO_3_ (SFC), and FeNO_3_@(BiO)_2_CO_3_ (SFB) were synthesized by coprecipitation. The purpose was to obtain materials that possess targeted adsorbent properties for the recovery of silver ions from aqueous solutions. After synthesis, to emphasize the adsorptive qualities of materials for the recovery of silver ions, the synthesized composite materials, as well as those doped with silver ions following the adsorption process (SFL-Ag, SFC-Ag, and SFB-Ag), were characterized and several adsorption-specific parameters were examined, including temperature, contact time, pH, adsorbent dose, and the initial concentration of silver ions in solution. Subsequently, the ideal adsorption conditions were determined to be as follows: pH > 4, contact time 60 min, temperature 298 K, and solid–liquid ratio (S–L) of 0.1 g of adsorbent to 25 mL of Ag (I) solution for all three materials. The Langmuir model properly fits the experimental equilibrium data of the adsorption process; however, the Ho–McKay model closely represents the adsorption kinetics. The maximum adsorption capacities of the materials, 19.7 mg Ag(I)/g for SFC, 19.3 mg Ag(I)/g for SFB, and 19.9 mg Ag(I)/g for SFL, are comparable. The adsorption mechanism is physical in nature, as evidenced by the activation energies of 1.6 kJ/mol for SFC, 4.15 kJ/mol for SFB, and 1.32 kJ/mol for SFL. The highest Ag(I) concentration used for doping all three materials in the study was 150 mg Ag(I)/L. The process is endothermic, spontaneous, and takes place at the interface between the adsorbent and the adsorbate, according to thermodynamic theory. Subsequently, the antimicrobial activity against *Staphylococcus aureus*, *Escherichia coli*, *Pseudomonas aeruginosa*, and *Candida albicans* microorganisms was evaluated by rate of inhibition assessment. The SFC-Ag material showed a percentage of 100% inhibition with respect to the positive control for each microorganism. All synthetized materials have better efficiency as antifungal agents.

## 1. Introduction

Silver, as a transition element, is one of the essential precious metals, found mainly in silver ores [1,2], whose market demand is increasing, and silver production is below the necessary [3]. Thus, the recovery of silver from secondary sources is urgently needed, as silver finds applications in various industries, the more important uses being in the electronic device field [2,3,4], in the renewable energy area, in solar panels [5,6,7], or in the medical field [8,9,10,11,12,13]. Despite the increased number of applications, silver’s natural resources have been depleted, and production costs have been rising quickly. In this sense, the recovery of silver from aqueous media by various methods, each with various advantages and disadvantages, is necessary. In the specialized literature, the most common methods used for silver recovery are chemical precipitation, solvent extraction [14,15,16,17], electrolysis [18] or ionic exchange [19,20,21], but also adsorption methods [22,23,24]. Among the commonly used methods, adsorption is a process to recover silver from solutions, and with increased efficiency, it is a reasonable alternative from an economic point of view. In the adsorption process, the most important property of an adsorbent material is its porous structure. The more porous the material, the greater the contact surface area and the greater the material’s ability to retain the metal ions. The adsorbent must have good mechanical properties, such as wear resistance, and must also be easily regenerated. Consequently, for the recovery of metal ions using materials with adsorbent properties, the choice is made based on the contact surface area and pore volume [25,26,27].

Various materials were studied as silver adsorbents, such as biochar, zeolites, silica gel, polymers, biological materials, or functionalized ceramic materials. Some of these are identified in Table 1, along with the adsorption capacity identified in the studies.

In this study, for the recovery of silver, three composite materials based on iron and copper, bismuth or lithium carbonates were synthesized by the coprecipitation method. Coprecipitation was chosen as the synthesis method because (i) it allows for intimate mixing of the metal precursors at the atomic or molecular level in solution. This leads to a much more uniform distribution of the components in the final product compared to the physical methods of mixing preformed oxides; (ii) it allows the formation of the desired compound at temperatures lower than those required in conventional solid-state reaction methods. This can lead to reduced energy consumption and to limiting excessive particle growth, favoring the obtaining of materials with nanometric dimensions or fine microstructures; (iii) it is, in general, a relatively simple and economical technique, which does not require sophisticated equipment compared to other advanced synthesis methods; (iv) it presents simplicity and low cost [26,42,43].

In addition, material homogeneity is crucial for optimal synergistic properties. Thus, the synthesis aimed to precisely adjust the concentrations of the precursor solutions, to precisely control the ratio of the different metal oxides in the resulting composite. This is essential to optimize the performance of the material for the desired application. Reaction parameters, such as pH, temperature, precursor concentration, and precipitant agent, were adjusted to influence the size, shape, and degree of agglomeration of the resulting composite particles [42,44].

Subsequently, the synthesized materials were used as adsorbent materials for the recovery of Ag(I) from aqueous solutions. The recovery process of silver ions from aqueous media resulted in depleted materials with a high silver content. These silver-doped materials can be used in various medical applications, as they exhibit significant antimicrobial activity due to the antimicrobial properties of silver ions [45,46,47].

Microorganisms found in their surroundings, including bacteria, fungi, and viruses, frequently infect humans. Due to the rise in the number of antibiotic-resistant microorganisms and the ongoing focus on health care costs, numerous scientists have studied ways to create new, efficient antimicrobial agents that are both economical and effective at combating these microorganisms’ resistances [48,49]. The usage of silver-based antiseptics, which may have a broad spectrum of action and a significantly lower propensity to cause microbial resistance than antibiotics, has increased as a result of these issues and demands. However, to date, the antimicrobial mechanisms of silver ions have long been debated and have not yet been fully elucidated. Most hypotheses consider silver ions to be the main active agents responsible for antimicrobial effects because they disrupt the selective barrier function of bacterial cell membranes, which causes increased permeability and facilitates the easy entry of toxic elements into the cell. Another hypothesis is that these ions can bind to essential cellular components, such as DNA and proteins, interfering with their function and causing denaturation. Metal ions are also considered to inhibit enzymes and disrupt the electron transport chain, leading to the production of reactive oxygen species (ROS) and cell damage [47,50,51,52].

These antimicrobial properties of silver-doped materials make them suitable for a wide range of applications, including medical devices and implants to prevent infections and biofilm formation [53,54], bandages and impregnated dressings to inhibit bacterial growth [55] and allow healing, wound dressings, textiles and personal care products [56], sanitary napkins and antimicrobial cosmetics. Silver-doped materials can also be used for water purification [54,57] or food packaging that can extend shelf life by inhibiting microbial growth [58,59,60]. Silver-doped materials can also be used in antimicrobial paints for medical facilities or for various self-disinfecting coatings and surfaces in public areas [61,62].

For all of these, the objectives of this study were as follows: (i) the synthesis of composite materials FeNO_3_@Li_2_CO_3_, (SFL), FeNO_3_@CuCO_3_ (SFC), and FeNO_3_@(BiO)_2_CO_3_, (SFB); (ii) the physicochemical characterization of the synthesis materials, but also of the materials doped with silver ions after adsorption (SFC-Ag, SFB-Ag, and SFL-Ag) by Fourier Transform Infra-Red spectroscopy (FT-IR), atomic force microscopy (AFM), and small angle X-ray diffraction (SAXS). Also, the point of zero charge (pH_pZc_) of the material was determined and the specific surface area was determined using the Brunauer–Emmett–Teller (BET) method; (iii) evaluation of the mechanism of Ag(I) recovery by adsorption, establishing the process mechanism through kinetic, thermodynamic, and equilibrium studies starting from the specific parameters of the adsorption process (adsorbent dose, pH, contact time, temperature, and initial concentration of Ag(I) in solution); (iv) application of Ag(I)-doped materials as antimicrobial agent.

Consequently, the study discusses a composite material, which is a combination of multiple components, rather than a single, specific chemical composition. This is significant because such a material would likely leverage the unique properties of each ingredient: carbonates for their adsorptive capabilities, bismuth and copper for their antimicrobial effects and potential roles in silver binding, and iron for possible magnetic separation or redox reactions. Its application in fields like bioremediation, biosorption, or the regulation of microbiological growth while recovering silver is suggested by its microbiological uses.

## 2. Materials and Methods


**Materials synthesis**


The coprecipitation method was used to obtain composites SFL, SFC, and SFB. In a solution obtained by dissolving 1 g of precursor (copper carbonate (II), 95%, extra pure, ROTH, Karlsruhe, Germany), bismuth carbonate (99.99% purity, Merck, Sigma Aldrich, Darmstadt, Germany), lithium carbonate (99.99% purity, Merck, Sigma Aldrich, Darmstadt, Germany) in 30 mL of distilled water, 30 mL of methanol (99.99% purity, Merck, Sigma Aldrich, Darmstadt, Germany) was added. The mixture was stirred for one hour. Then, 5 mL of HNO_3_ solution (Merck, Sigma Aldrich, Darmstadt, Germany) with a pH ~2 was added to the initial solution and stirred for another 30 min until the metal carbonate dissolved. Next, 5 g of Fe(NO_3_)_3_ (Merck, Sigma Aldrich, Darmstadt, Germany) were added to the reaction mixture, and the temperature was raised to 50 °C while stirring continued for 3 h until homogenization was achieved. To induce precipitation, 10 mL of NaOH solution (prepared by dissolving 7.5 g of NaOH in 100 mL of distilled water), (Merck, Sigma Aldrich, Darmstadt, Germany), was rapidly added to the reaction mixture. The precipitate was separated and thoroughly washed with excess distilled water. The final product, in powder form, was dried in an oven (Pol-Eko model SLW 53, SDT Poland) at 100 °C for 24 h.


**Material Characterization**



**Fourier Transform Infra-Red spectroscopy, FT-IR**


The materials were characterized by FT-IR spectroscopy, conducted at room temperature using a Nicolet™ iS50 FTIR Spectrometer (Donau, Austria). IR spectra were recorded in the spectral range of 4000–400 cm^−1^, with 64 scans at a resolution of 16 cm^−1^.


**Atomic Force Microscopy, AFM**


The synthesized material, dried at 100 °C, was analyzed by atomic force microscopy. AFM images were obtained using a Scanning Probe Microscopy Platform (MultiView-2000 system, Nanonics Imaging Ltd., Jerusalem, Israel) operating in intermittent mode under standard conditions (298 K). The analysis was performed with a chromium-doped tip having a radius of 20 nm and a resonance frequency of 30–40 kHz.


**Small Angle X-ray Diffraction, SAXS**


Small-Angle X-ray Scattering measurements were performed using a Xenocs Xeuss 3.0 system (Xenocs SAS, Grenoble, France). Measurements were carried out under vacuum at room temperature, and the samples were mounted in a powder holder using Kapton film and calibrated with silver behenate. The detector (Dectris) was positioned at a distance of 1.8 m, and measurements were conducted in very high flux mode using a Cu radiation source. Data fitting was performed using the Porod model in the q-range of 0.1–1 nm^−1^ for all samples. The goodness-of-fit (χ^2^) was approximately 1.5 for all cases.


**Porosity and surface area measurements**


Nitrogen desorption isotherms were measured at 77 K using a Nova 1200e instrument (Quantachrome Instruments, Boynton Beach, FL, USA). Pore size distributions of the pore were calculated using the Barrett–Joyner–Halenda (BJH) method and specific surface areas were determined using the Brunauer–Emmett–Teller (BET) method. Before analysis, the samples were degassed under vacuum for 5 h at room temperature. Surface area measurements for various synthesized batches were repeated under identical conditions, and the results showed values within a standard error of 5%.


**The zero charge point, pH_pZc_, of the material**


Also, the zero charge point was determined by the method of bringing the system studied to equilibrium. An amount of 0.1 g of material was used for this study, which was mixed with 25 mL of 0.1 N KCl solution (Carl Roth GmbH + Co, Karlsruhe, Germany) at 200 rpm and a temperature of 298 K, using a water bath with thermostating and stirring (Julabo SW23, Seelbach, Germany). The pH of the KCl solutions was adjusted in the range of 1–14, using the NaOH solution (Carl Roth GmbH + Co, Karlsruhe, Germany) with a concentration between 0.05 N and 2N or the solution of HNO_3_ (Carl Roth GmbH + Co, Karlsruhe, Germany) with a concentration between 0.05 N and 2 N. The samples were filtered and then the pH of the resulting solution was determined using a pH meter (Mettler Toledo, SevenCompact, S 210, Schwerzenbach, Switzerland).


**Adsorption studies**



**Material dose effect**


To establish what is the effect of the dose of the material to obtain the best efficiency of the adsorption process, different amounts of each material studied (0.05, 0.1, 0.15, 0.2, 0.25 and 0.3 g of material) were weighed for a constant volume (25 mL) of Ag(I) (AgNO_3_ pa, Merck, Darmstadt, Germany), concentration of 10 mg/L.

Adsorptions were carried out using a shaker (Julabo SW23, Seelbach, Germany), at a temperature of 298 K, for a contact time of 60 min. The adsorption efficiency was determined using the following relation:(1)$$Efficiency\;[\%]=\frac{C_{i}-\;C_{rez}}{C_{i}}\times 100$$
where

$C_{i}$—Ag(I) initial concentration (mg/L)

$C_{rez}$—Ag(I) residual concentration (mg/L)


**pH effect**


The influence of the pH of the solution was studied in the pH range of 1 to 14. A total of 0.1 g of adsorbent material was mixed with 25 mL of Ag(I) solution with an initial concentration of 10 mg/L. Contact time was 60 min at temperature 298 K. The pH of the solutions was adjusted using HCl or NaOH solutions with concentrations between 0.1 and 1 N.

The material adsorption capacity was evaluated using the following relation:(2)$$q\;\left[\frac{mg}{g}\right]=\frac{\left(c_{i}-c_{rez}\right)\times V}{m}$$
where

q—adsorption capacity (mg/g)

$C_{i}$—Ag(I) initial concentration (mg/L)

$C_{rez}$—Ag(I) residual concentration (mg/L)

V—solution volume (L)

m—material mass (g)


**Effect of contact time and temperature**


To establish the influence of contact time and temperature over adsorption capacity, samples of 0.1 g were weighed, which were mixed with 25 mL solutions of Ag(I), with an initial concentration of 10 mg/L. The samples obtained were mixed at 200 rpm for different time periods (15, 30, 45, 60, 90, and 120 min) at different temperatures (298, 308, 318, and 328 K).


**Initial concentration effect**


To evaluate the effect of the initial concentration of Ag(I) ions on the maximum adsorption capacity, solutions with initial concentrations of 5, 10, 20, 50, 100, 130, 150, 160, and 170 mg/L were prepared. All adsorptions were carried out in pH > 4, at 298 K for a mixing time of 60 min, according to the initial step of the pH effect, the temperature and the effect of contact time.


**Adsorption mechanism**


To establish the mechanism of the adsorption process, kinetic, thermodynamic, and equilibrium studies were performed.


**Adsorption isotherms**


Three models of equilibrium isotherms were used, namely: (i) the Langmuir isotherm which is based on the monolayer adsorption of the solute; (ii) the Freundlich isotherm, which was originally developed for heterogeneous surfaces, and (iii) the Sips model, which is a model that combines the Langmuir and Freundlich models.

The equilibrium isotherms were obtained by graphical representation of the *q_e_* = *f*(*Ce*), and the related parameters were obtained for each isotherm. The Langmuir and Freundlich constants were calculated using the linearized form of the resulting patterns after fitting.

The Langmuir isotherm is [63](3)$$q_{e}=\frac{q_{L}K_{L}C_{e}}{1+K_{L}C_{e}}$$
where

qe—equilibrium adsorption capacity (mg/g)

Ce—equilibrium concentration (mg/L)

$q_{L}$—Langmuir maximum adsorption capacity (mg/g)

$K_{L}$—Langmuir constant.

The Freundlich isotherm is [64](4)qe=KFCe1nF
where

$q_{e}$—maximum adsorption capacity (mg/g)

Ce—equilibrium concentration (mg/L)

$K_{F}$ and $n_{F}$—characteristic constants that can be associated with the relative adsorption capacity of the adsorbent and the adsorption intensity.

The Sips isotherm is [65](5)qe=qSKSCe1nS1+KSCe1nS
where

$q_{S}$—Sips maximum adsorption capacity (mg/g)

$K_{S}$—constant related to adsorbent adsorption capacity

$n_{S}$—heterogeneity factor.


**Adsorption kinetics**


The pseudo-first-order kinetic models, the Lagergren model, and pseudo-second-order kinetic models, or the Ho and McKay models, were applied to describe the kinetics of the adsorption of Ag(I) during the process.

The Lagergren model is [66](6)$$\ln(q_{e}-q_{t})=lnq_{e}-k_{1}t$$
where

qe—equilibrium adsorption capacity (mg/g)

qt—adsorption capacity at a specific time *t* (mg/g)

$k_{1}$—pseudo-first-order speed constant (min^−1^)

t—contact time (min)

The Ho and McKay model is [67](7)$$\frac{t}{q_{t}}=\frac{1}{k_{2q_{e}^{2}}}+\frac{t}{q_{e}}$$
where

$q_{e}$—equilibrium absorption capacity (mg/g)

qt—adsorption capacity at a specific time *t* (mg/g)

$k_{2}$—pseudo-second-order speed constant (g/mg·min)

t—contact time (min)

A linear plot of *ln*(*qe* − *qt*) = *f*(*t*) gives the equilibrium adsorption capacity, *qe_,calc_*, as intercept, and the slope gives the rate constant *k*_1_ for the pseudo-first-order kinetic model; a linear plot of *t*/*qt* = *f*(*t*) gives the slope as equilibrium adsorption capacity *qe,_calc_,* and the intercept gives the rate constant *k*_2_ for the pseudo-second-order kinetic model.

The intraparticle diffusion represents the movement of adsorbate species within the pores of the adsorbent material after their initial attachment to the surface. It is one of the key steps in the adsorption process along with film diffusion (the transport of adsorbate from the bulk solution to the surface of the adsorbent) and surface adsorption (the interaction of the adsorbate with the adsorbent surface). The intraparticle diffusion was described by the Weber and Morris model. The Weber and Morris model is [68](8)$$q_{t}=k_{diff}\times {\;t}^{1/2}+C$$
where

$q_{t}$—adsorption capacity at t time (mg/g)

$k_{diff}$—speed constant for intraparticle diffusion (mg/g·min^1/2^)

C—constant parameter correlated with the thickness of the liquid film surrounding the adsorbent particles.

The activation energy, *E_a_*, can give information on the nature of the adsorption process, be it physical or chemical. Thus, the activation energy was calculated using the Arrhenius equation and the velocity constant from the pseudo-order kinetic model two, *k*_2_.(9)$$k_{2}=Ae^{\frac{-E_{a}}{RT}}$$
where

$k_{2}$—rate constant (g/min·mg)

A—Arrhenius constant (g·min/mg)

$E_{a}$—activation energy (kJ/mol)

T—absolute temperature (K)

R—the ideal gas constant (8.314 J/mol·K)

The activation energy of the adsorption Ag(I) on the material was calculated from the equation of the line of the graphical representation *ln k*_2_ = *f*(1/*T*).


**Thermodynamics of the adsorption process**


The thermodynamic studies offer general information on the influence of temperature on the adsorption, and the results suggest the feasibility of the adsorption process. To determine how the adsorption of Ag(I) ions proceeds on the surface of the adsorbent, the Gibbs free energy (G°) was calculated using the Gibbs–Helmholtz equation [69]:(10)$$∆G^{0}=∆H^{0}-T\times ∆S^{0}$$
where

$∆G^{0}$—free Gibbs energy standard variation (J/mol)

$∆H^{0}$—enthalpy standard variation (J/mol)

$∆S^{0}$—entropy standard variation (J/mol·K)

T—absolute temperature (K)

Standard variations in enthalpy and entropy were evaluated from the linear dependence of *lnKd* = *f*(1/*T*) (linear form of the van’t Hoff equation), where *Kd* is the equilibrium constant, being calculated as the ratio between equilibrium adsorption capacity *(qe)* and equilibrium concentration ($C_{e}$).(11)$$lnKd=\frac{∆S^{o}}{R}-\frac{∆H^{o}}{RT}$$
where

Kd—constant of equilibrium

$∆S^{o}$—standard variation in entropy (J/mol·K)

$∆H^{o}$—standard variation in enthalpy (kJ/mol)

T—absolute temperature (K)

R—ideal gas constant (8.314 J/mol·K)

The equilibrium constant is given by the ratio of the *qe* and *Ce*:(12)$$Kd=\frac{q_{e}}{C_{e}}$$


**Microbiological activity**



**Bacterial culture preparation to evaluate the antimicrobial activity of synthetic compounds against reference strains**


Four reference strains from the American Type Culture Collection (Microbiologics, St. Cloud, MN, USA) namely *Staphylococcus aureus* ATCC 25923, *Pseudomonas aeruginosa* ATCC 27853, *Escherichia coli* ATCC 25922, and *Candida albicans* ATCC 10231 were used in microbiological control experiments to evaluate the synthetic materials’ antimicrobial activity against the reference strains. The optical density of the microbial inoculum was set to 0.5 McFarland (10^8^ CFU/mL). Ten microliters of the inoculum from each microbial strain was taken and placed on the Petri plate in which the test material was also incorporated. Each of the reference microbial strains were also inoculated into a control plate that contained only the microbial inoculum (10^6^ CFU/mL). The microbial strains were incubated at 37 °C, for 24 h. The quantitative growth of the microbial colonies after incubation was assessed by determining the total number of microbial colonies that developed on the culture medium (CFU/mL). The efficiency of the Ag(I)-doped material on reference strains was expressed as the rate of inhibition of bacterial growth. The growth inhibition ratio of microbial strains was expressed as a percentage, according to the equation:(13)$$inhibition\;rate=\;\frac{{CFU}_{control}-{CFU}_{test}}{{CFU}_{control}}\times 100$$
where

${CFU}_{control}$ = the number of colonies on the control plate.

${CFU}_{test}$ = the number of colonies on the test plate.

## 3. Results and Discussion


**Materials synthesis and characterization**



**FT-IR spectra**


In Figure 1, FT-IR spectroscopy is presented for the synthesized samples.

The FTIR spectrum of the SFL, SFC, and SFB materials shows the presence of the characteristic strong bands corresponding to the ${CO}_{3}^{2-}$ anion located at ~1600–1400 cm^−1^ (asymmetrical stretching C-O) together with weaker bands at 1100–1000 cm^−1^ (symmetric stretching vibrations), and 850–800 cm^−1^ (bending out-of-plane vibrations) [70,71,72,73,74].

The band observed at 1400–1300 cm^−1^ corresponds to the asymmetric stretching vibration of the ${NO}_{3}^{-}$ group, and the band at 805 cm^−1^ corresponds to the bending out-of-plane vibration of the ${NO}_{3}^{-}$ group [75,76,77,78].

The 600–200 cm^−1^ region is attributed to the M-O bonds, which confirms the presence of Cu-O, Li-O, Bi-O, and Fe-O bonds [74,79,80].

In the range of 600–400 cm^−1^, specific vibrations of the Cu-O bond can appear [72,78], bonds that can be composed of Cu_2_O or CuO compounds. In the case of bismuth oxide (Bi_2_O_3_), specific bands of the Bi-O bond can appear in the range 550–300 cm^−1^ [73,74]. Li-O bonds appear in the range 50–350 cm^−1^ [71]. In particular, the band around 500 cm^−1^ is assigned to Fe–O stretching vibrations, in agreement with the literature data [75,79,80].


**Atomic force microscopy, AFM**


AFM images of the SFL, SFC, and SFB materials together with their roughness data obtained from the AFM images (average roughness (*S_a_*), mean square root roughness (*S_q_*), maximum peak height (*S_p_*), maximum valley depth (*S_v_*), maximum peak-to-valley height (*S_y_*), surface kurtosis (*S_ku_*) and surface skewness (*S_sk_*)) are presented in Figure 2, and Table 2, respectively.

SFC material has the highest roughness values with *S_a_* = 221 and *S_q_* = 287, followed by the SFB with *S_a_* = 193.7 and *S_q_* = 251. Lowest roughness values are in the case of SFL with S_a_ = 119.8 and *S_q_
*= 166. The same tendency is observed in the case of *S_p_* and *S_v_*, and *S_y_* values, respectively. However, the SFL lowers *S_p_* and *S_v_*, and *S_y_* values, which may be the result of the material’s morphology. This feature is also observed in the AFM images, whereas the SFL clusters are scattered on the surface at great distances, leaving a lot of free space between them. In the case of SFC and SFB, materials exhibit different morphologies; however, both surfaces are well packed. In the case of all samples, the *S_ku_* > 3 value reveals many high peaks and a low valley together with *S_sk_* ≈ 0 specific for symmetrical height distribution whereas neither peaks nor valleys are dominant [81]. Both *S_ku_* and *S_sk_* values are the highest in the case of SFL, probably due to the scattered arrangement of the material on the surface.


**Measurements of porosity and surface area**


Figure 3 shows the nitrogen adsorption–desorption isotherms and the corresponding pore size distribution (inset) for the synthesized samples.

Analyzing the results obtained and comparing with IUPAC [82], it can be concluded that the nitrogen adsorption–desorption isotherms are type IVa with H3 hysteresis typical for mesoporous materials with non-rigid aggregates of plate-like particles.

By comparing the samples (see Table 3), it can be seen from the below data of the isotherms that the specific surface area is influenced by the type of carbonate salt used.

The materials used for synthesis in composites with different atomic radiuses (Li (152 pm), Cu (128 pm), Bi (160 pm)) indicate a significant influence on the structure and texture of the material, observing the same trend in the case of the results obtained from porosity and fractality. When the pore size from the isotherms and those obtained from the average roughness (S_a_) are analyzed, a similarity can be observed, confirming the authenticity of the results. The fractal parameters obtained from the FHH method indicate a 2D surface for all samples.


**Small Angle X-ray Diffraction**


Detailed data extracted from the SAXS region provide valuable information on the structure and morphology of the materials. Figure 4 presents the synthesized materials using various metal carbonates. To enable comparison, the fitting was performed within the same Porod region.(14)$$l\left(q\right)=B+K\times q^{-D}$$
where

B = background line,

K = Porod slope

D= related to the fractal dimension of the scattering interface.

The results obtained are shown in Table 4.

The results obtained from the scattering data in the small q region 0.1 nm^−1^ and 1 nm^−1^ indicate that all values are less than 3 for the fractal dimension. Values indicate that the materials are characterized by mass fractals [83]. Additionally, the shape of the nanoparticles indicates the behavior of the plate with a rough surface, which can be observed from the adsorption–desorption isotherms of N_2_ [84,85].

AFM provides images of the sample’s topography in real space, while SAXS offers statistical, average information about size, shape, and aggregation states from a larger sample volume in reciprocal space. Since the AFM-obtained data refers only to the scanned area (with the maximum possibility of scanning a 50 × 50 μm area), some differences in the case of SAXS data can be expected. The correlation lies in how they provide different but consistent perspectives, with AFM’s real-space visualizations confirming SAXS data and model interpretation, thus increasing the reliability and providing a more complete understanding of the sample’s overall and local structure. The discrepancy between the SAXS and AFM data is common as suggested by Christopher K. et al. [86].

Although the median data obtained from the SAXS and the observed dimensions in AFM images are not similar for the same reason explained above, when it comes to the different method of measurement whereas AFM is direct while SAXS generates data in reciprocal space, requiring modeling, Fourier transformations, or fitting to obtain real-space structural information, the fractal dimension data obtained from SAXS and surface kurtosis data confirm the prevalent rough surface with high peaks and low valleys.


**The zero-point charge, pH_pZc_**


The pH_pZc_ value associated with the synthesized materials was obtained by graphically representing the final pH value (pH_f_) versus the initial pH value (pH_i_) (Figure 5).

Knowledge of the acidic–basic properties of materials with adsorbent properties plays an important role in their use. The point of zero charge represents the appearance of a potential at the interface of the system, owing to the existence of H^+^/HO^−^ ion pairs. Under this condition, the pH_pZc_ was determined to obtain information about the nature of the electrical charge of the surface of the material. If pH > pH_pZc_, the material’s surface will have a negative charge, and if the pH < pH_pZc_, the material’s surface will be positive charging [87].

From Figure 5, it can be observed that the zero charge point of the SFL material is 10.4, or the SFC material is 5.5, and for the SFB material, it is 10.7. The species of silver ions that can be attracted to the surface of the materials, regardless of the material, is Ag^+^ [88].


**Silver Adsorption Processes**



**Adsorbent dose, pH, contact time, temperature, and initial concentration effect**


Specific parameters of the adsorption process are factors that influence the ability of an adsorbent material to retain the molecules of an adsorbate. These parameters can be classified according to the nature of the adsorbent (specific surface area, pore volume and distribution, chemical nature of the surface, morphology, and particle size), the nature of the adsorbate (size and shape of the molecules, polarity and electrical charge, concentration of the adsorbate) and also the operating conditions (pH, temperature, contact time).

Figure 6a–c,d1–d3 shows the effect of adsorption dose, pH, contact time, temperature and initial concentration of Ag(I) ions on the efficiency of the adsorption process, and on the adsorption capacity of the materials under study, respectively.

Generally, a larger amount of adsorbent material available in the system will result in a higher adsorption capacity. This is due to the presence of a larger number of active sites on the adsorbent surface, which are available to bind adsorbate molecules. As the adsorbent dose increases, the amount of adsorbate retained will increase until a saturation point is reached. Beyond this optimal dose, further increases in the amount of adsorbent will not result in a significant increase in the amount of adsorbate removed from the fluid phase. This is because all of the available adsorbate is already bound to the adsorbent surface.

Similarly, it is observed in Figure 6a, when, regardless of the nature of the adsorbent, at ratios higher than 0.1 g adsorbent:25 mL Ag(I) solution, the efficiency of the adsorption process remains approximately constant. For example, in the case of using SFL material, the efficiency is lower, but constant, 79%, in the case of using SFC material the efficiency is the highest, 90%, and in the case of using the SFB material, the efficiency is 87%. Thus, subsequent studies were carried out at the ratio S–L = 0.1 g:25 mL.

It is known that pH is the control parameter associated with the adsorption process and can be influenced by the ionic form of Ag(I), but also by the nature of the functional groups existing on the surface of the synthesized materials. Because of this fact, the optimal pH for the recovery of Ag(I) from aqueous solutions by adsorption was established. Experimental data (Figure 6b) indicate that, with increasing pH, the adsorption capacity of the material increases, regardless of the nature of the adsorbent, so that the adsorption process proceeds with good results at acidic pH, (pH > 4). At this pH, the species of Ag(I) ions found in the solution, according to data from the literature, is Ag^+^ [89,90].

It can be observed that for the SFC material, the adsorption capacity is higher (*q* = ~2.22 mg/g) compared to the SFL material which has the lowest adsorption capacity (*q* = ~2 mg/g).

The concentration of Ag(I) ions in the solution is a key factor affecting the adsorption performance. The adsorption performance for different initial concentrations of Ag(I) ions was studied for all materials (SFL, SFC, and SFB), as shown in Figure 6c The adsorption capacity initially increases rapidly with increasing Ag(I) concentration, followed by a gradual slowdown as equilibrium is reached. This trend was attributed to the higher concentration gradient at the initial stage of adsorption, which increased the driving force for adsorption and led to an increase in the adsorption capacity. At lower initial concentrations, the adsorption sites on the surface of any material are abundant and readily available. When the initial concentration of Ag(I) increases, the adsorption capacity increases rapidly. However, as the concentration continues to increase, a multitude of adsorption sites are occupied, and the available adsorption sites on the surface of any material begin to be saturated, which leads to a slowdown in the rate of increase in the adsorption capacity. Finally, at high Ag(I) concentrations, the adsorption capacity stabilizes and approaches equilibrium because most of the active sites are saturated, which reflects the maximum adsorption capacity achievable under the given conditions. The maximum adsorption capacities of all the materials were as follows: 19.9 mg Ag(I)/g for SFL, 19.7 mg Ag(I)/g for SFC, and 19.3 mg Ag(I)/g for SFB for the same initial concentration, *C_i_
*= 150 mg Ag(I)/L.

It can be seen from Figure 6(d1–d3) that with increasing contact time, the adsorption capacity of the materials increases, regardless of the nature of the adsorbent. After 60 min, the adsorption capacity remains constant (between 1.99 mg Ag(I)/g and 2.17 mg Ag(I)/g). Additionally, with increasing temperature, the adsorption capacity of the material also increases. Since the increase in adsorption capacity is not significant with increasing temperature, further studies were carried out at 298 K.


**Adsorption Mechanism**


Specific parameters for modeling the adsorption process are (i) the maximum adsorption capacity: the maximum amount of adsorbate that can be retained by a unit amount of adsorbent under saturation conditions; (ii) the equilibrium constant of the adsorption that reflects the affinity of the adsorbent for the adsorbate; a higher value indicates a stronger affinity; (iii) factors from isotherm equations that characterize the nature of interactions and the heterogeneity of the surface, (iv) kinetic parameters that describe the rate at which adsorption occurs; (v) thermodynamic parameters (Δ*H*, Δ*S*, Δ*G*) that provide information about the energy and spontaneity of the adsorption process.


**Kinetic studies**


In order to analyze the kinetics of the Ag(I) adsorption process, but also to understand the kinetic mechanism governing the adsorption process, the experimental data obtained were modeled using three different kinetic models, namely: the pseudo-first-order kinetic model (Lagergren model), the pseudo-second-order kinetic model (Ho–McKay model), and the intraparticle diffusion (Weber and Morris model), at three working temperatures (298, 308, and 318 K).

The isotherms obtained for the three models for the three synthesized materials are presented in Figure 7a–i.

The specific kinetic parameters, for all the synthesized materials, at the studied temperatures are presented in Table 5.

The rate constants *k*_1_ and *k*_2_ of the pseudo-first-order and pseudo-second-order kinetic models, as well as the adsorption capacities *q_e_,_calc_*, were obtained for each synthesized material using the line equations *ln*(*qe* − *qt*) = *f*(*t*) and *t*/*qt* = *f*(*t*). It was feasible to determine that the model accurately depicts the adsorption process of Ag(I) on the materials based on the kinetic parameters obtained for each model and the correlation coefficient, *R*^2^ values. According to the *R*^2^ values, the experimental data acquired for all materials under study can be described by the pseudo-second-order kinetic model.

The experimental kinetic data were analyzed using the Weber–Morris model to determine whether film diffusion or intraparticle diffusion is the rate-determining step. By determining whether the process occurs in multiple steps, the mechanism of Ag(I) adsorption can be determined. Plotting *qt* = *f*(*t*1/2) at the three operating temperatures allowed for the establishment of this. After calculating the parameters *K_diff_* and *C*, it was determined whether or not the resulting straight lines pass through the origin.

Furthermore, it can be shown from the data in Table 5 that the value of K_diff_ increases as the temperature for all materials increases. Since stage 1’s diffusion constants are greater than stage 2’s, we can conclude that stage 2 determines the rate at which the process proceeds [91].


**Thermodynamic studies**


Thermodynamic investigations were conducted in the temperature range of 298–318 K to obtain information on the energy changes related to the adsorption process. The variations in free entropy and free enthalpy of the three materials were calculated using the linear formulation of the dependence *lnKd* = *f*(1/*T*) (Figure 8a). The fluctuation of Gibbs free energy was then assessed using the van’t Hoff equation.

The minimal energy needed to convert the reactants into reaction products must be determined, since chemical reactions are crucial in regulating the rate of the adsorption process. Therefore, using the value of the rate constant *k*_2_ and the Arrhenius equation, the activation energy needed for the adsorption of Ag(I) ions was determined for the three materials (Figure 8b).

Table 6 presents the thermodynamic parameters specific to the adsorption process, and Table 7 presents the activation energy for the three materials studied.

The Gibbs free energy (Δ*G*), enthalpy change (Δ*H*), and entropy change (Δ*S*) were measured in order to fully evaluate the thermodynamic characteristics of Ag(I) adsorption on the three materials. It is evident from Table 6’s results of the specific thermodynamic parameters that the adsorption process is endothermic. Since the enthalpy change (Δ*H*) is positive, Ag(I) adsorption is favored at high temperatures. The adsorption reaction is spontaneous under the studied conditions, and its spontaneity increases with temperature, as confirmed by the fact that all the observed Δ*G* values are negative. Furthermore, the increased unpredictability of the solid–liquid interface is reflected in the positive change in entropy change (Δ*S*). According to this thermodynamic analysis, Ag(I) adsorption on any of the SFL, SFC, or SFB materials is an endothermic process. As the temperature increases, the equilibrium shifts in a positive direction, increasing the adsorption capacity.

The minimal amount of energy needed to start a process is known as the activation energy, Ea. The Arrhenius equation states that temperature affects reaction rate, meaning that a higher temperature causes the process to proceed more quickly [92]. In the scenario under study, a few collisions inevitably happen when the adsorbent and the adsorbate come into contact. This is particularly valid in cases where the kinetic energy is minimal. The free energy of the system must be exceeded in order to adsorb the Ag(I) ions. Because the process is endotherm, the heat from the surroundings provides the activation energy that is required. The molecules move faster, increasing the probability of contact and the forces of interaction [93]. The activation energy values can provide information about the nature of the adsorption process. It is found that the recovery by adsorption of Ag(I) on the three materials is of physical or physicochemical nature [92].


**Equilibrium studies. Adsorption isotherms**


The interaction between the solution and the adsorbent material must be described in order to determine the mechanism by which adsorption takes place. Equilibrium isotherms, which illustrate the correlation between the concentration of Ag(I) ions remaining in the aqueous phase (*C_e_*) and the amount of material adsorbed per gram of adsorbent at equilibrium (*q_e_*), can be utilized to achieve this. Three adsorption isotherms were used, the Langmuir, Freundlich, and Sips isotherms, to mathematically model the experimental data in order to describe the Ag(I) adsorption process on the three manufactured materials (Figure 9).

The isotherm parameters obtained through mathematical modeling are presented in Table 8.

It is evident from the data in Table 8 that the Langmuir isotherm has the correlation coefficient, *R*^2^, which is closest to 1 regardless of the material used for the recovery of Ag(I). This fact allows us to state that the experimental data correlate best according to this isotherm. Furthermore, it is observed that the adsorption capacity values obtained by modeling the data using the Langmuir isotherm (23–25 mg/g) are closer to those obtained experimentally (19.3–19.9 mg/g). This helps to explain why this isotherm was chosen as the one that best describes the adsorption process. Essentially, it can be said that the adsorption process of Ag(I) on any of the three produced materials occurs on monolayers, as the experimental results are modeled with an excellent regression coefficient according to the Langmuir model. Ag(I) molecules form a single layer on the surface after binding. No further Ag(I) molecules can be adsorbed on an adsorption site once it is occupied. Additionally, every adsorption site on the surface of the adsorbent is the same and has the same adsorption energy [69].


**Materials characterization post-doping with silver**


These studies require a comprehensive analysis of the properties of the material after silver has been added, by adsorption.


**Fourier-transform infra-red sp**
**ectra after silver doping**


After the Ag(I) addition, the spectra of all materials changed, see Figure 10.

For all three materials, the carbonate-specific band at 1600–1100 cm^−1^ became weaker. The N-O band is more intense than the C-O band, but the overlap of the specific C-O and N-O peaks in the area 1350–1410 cm^−1^ is even more evident after the Ag(I) addition. The same happens in the 800–860 cm^−1^ region, where the C-O and N-O bands are present, revealing a more intense N-O band after the Ag(I) adsorption.

Instead, it can be observed that the specific M-O bands of the materials intensify, and at the same time, the specific Ag-O bands observed at ~550, 450, and 180 cm^−1^ also appear [94,95].


**Atomic force microscopy after silver-doping**


Figure 11 shows the AFM images obtained for 2D and 3D for synthesized materials after silver doping.

Compared to images before Ag(I) doping, all three materials are subjected to morphological changes, see Table 9.

A more dispersed surface was noticed in the case of SFC-Ag and SFB-Ag materials, whereas in the case of SFL-Ag, the opposite occurred. Given the results, it is clear that Ag had influence on the rugosity parameters of the materials, ensuing increased roughness values in the case of SFL-Ag and decreased roughness values in the case of SFC-Ag and SFB-Ag. Another visible change in the parameters is the *S_ku_* value, which was the greatest in the case of SFL, whereas after the Ag introduction, it decreased significantly, denoting a tendency toward a surface with less high peaks and low valleys. However, in the case of SFC-Ag and SFB-Ag, *S_ku_* slightly increases. The *S_sk_* value in the case of SFL-Ag and SFC-Ag slightly decreases, respectively, and increases in the case of SFB-Ag; still, their values are around zero, maintaining their symmetrical height distribution.


**Porosity and surface area measurements after silver-doping**


In Figure 12, the nitrogen adsorption–desorption isotherms are presented and show the average pore distribution for the materials after silver adsorption has occurred in the material.

The nitrogen adsorption–desorption isotherms obtained and compared with IUPAC data show that the SFL-Ag sample presents a type IVa isotherm but with a type H2(a) hysteresis that can be attributed to pore blocking in a narrow range of pore necks. In the case of the SFC-Ag sample, hysteresis is of type H2(b) specific for pore blocking, but the size of the pore neck is much larger. In the case of the SFB-Ag sample, the isotherm is also of type IVa with a type H3 hysteresis. Table 10 presents experimental data resulting from nitrogen adsorption–desorption isotherms for silver-doped materials.

The data presented indicates that in the case of the SFL-Ag and SFC-Ag samples, important changes in morphology occur, observed both from the specific surfaces and from the pore size. In the case of the SFB-Ag sample, it is observed that surface bonds are formed with the silver particles, observed by a decrease in the specific surface area and total pore volume compared to the initial ones, also observed from the AFM data.


**SAXS with silver-doping**


In Figure 13 the measurements and the fitted curves, in the *q* = 0.1 nm^−1^ and 1 nm^−1^, for SAXS samples doped with silver are presented.

The results obtained are shown in Table 11.

The fractal dimension extracted from the Porod domain indicates values above 3 for sample SFL-Ag and SFC-Ag observed also from the FHH method being specific for surface fractals, meaning that the sample indicates a more complex roughness shown also in AFM data.

In case of sample SFB-Ag, the fractal dimension is similar in the FHH method, indicating a mass fractality, showing that the roughness remains similar to the initial samples, but the microporosity increased as observed in all data having a value of 66 which is 2 times bigger than all the samples. These results showed that this material presents the best chance to be used in antimicrobial activity [96].


**Antimicrobial activity of iron–carbonate (Li, Cu, Bi) composites doped with silver**


The antimicrobial activity of iron–carbonate composites (Li, Cu, Bi) doped with Ag (I) are presented in Table 12.

The very good antimicrobial character of the SFC-Ag material seems to be given by the very large contact surface of the material with the microbial cell. SFC-Ag has a contact surface almost 10 times larger (53.1 m^2^/g) than SFB-Ag (4.9 m^2^/g), which determines a close contact of the material with the microbial wall, having as an effect the alteration of the cell permeability. Consequently, the microbial cell suffers a change in the capacity of the cell membrane to allow or inhibit the selective passage of toxic ions (Ag^+^), thus affecting the homeostatic balance of the cell and implicitly the microbial viability. As a result, there is a loss of control over molecular transport and by affecting the microbial cellular function, over time the cell destruction occurs as well as the inhibition of microbial proliferation, respectively. Therefore, this results in the very good antimicrobial effect of the synthesized SFC-Ag material.

Based on the same hypothesis, in the case of the SFB-Ag material, the contact surface is very low (4.9 m^2^/g), and exhibits reduced microbial inhibition, regardless of the concentration of doped silver ions (c = 1–50 mg/L Ag). Consequently, the rate of inhibition of microbial growth is much reduced (20–30%) compared to that obtained at the same concentration of doped Ag ions (100%). At the same time, in the case of the SFB-Ag material, the low antibacterial effect occurs, regardless of the tested strain. These are correlated with the morphology of the material, in the form of well-packed clusters, which do not allow strong contact, and occur on a large surface with the bacterial cell. This fact determines the reduction in the antibacterial effect of this type of material.

In the case of *S. aureus*, for the SFL-Ag material, obtaining an antibacterial effect inversely proportional to the concentration of doped Ag ions can be attributed to the homeostasis of Ag ions, corelated with the morphology of the material. The form of scattered clusters ensures good coverage of the bacterial cell surface. So, the synthesized materials tested respond well in terms of the observed antibacterial effect. It is possible that the peptidoglycan layer in the cell wall of Gram-negative bacteria aims for the contact with Ag ions on the surface of the doped materials, due to the difference in electronic charges. Thus, an increased influx of Ag ions occurs inside the bacterial cell, while cellular destruction occurs and implicitly the increase in the bacterial inhibition rate under the effect of the tested materials.

Regarding the antifungal character of the synthesized materials doped with silver ions, it should be emphasized that the synthesis of such materials with antifungal properties is desirable in the context in which 90% of infections are caused by Candida species, which are the most prevalent human fungal pathogens that induce cutaneous and mucosal or systemic infections [50].

The close contact between the tested material and the large contact surface for SFL-Ag (30.9 m^2^/g), and for SFC-Ag (53.1 m^2^/g), respectively, compared to the very low contact surface in the case of SFB-Ag (4.9 m^2^/g), result in a directly proportional antifungal effect.

## 4. Conclusions

In this study, to selectively recover Ag(I) from aqueous solutions, three adsorbents were created via coprecipitation. The Ag(I)-doped materials were subsequently employed as antimicrobial materials. Therefore, instead of referring to a single particular chemical compound, the study discusses a composite material, which is a mixture of various components. This is important because such a material would likely leverage the unique properties of each ingredient: iron for potential magnetic separation or redox reactions, bismuth and copper for their antimicrobial effects and possible roles in silver binding, and carbonates for their adsorptive capabilities. The microbiological purposes suggest its use in areas such as bioremediation, biosorption, or control of microbial growth while recovering silver.

Comprehensive characterization confirmed the synthesis of composites based on bismuth, copper, lithium, and iron oxides, the formation of mesoporous structures, and the increased physical–chemical stability of the materials.

The adsorption technique is thought to be appropriate for the recovery of beneficial substances or the purification of wastewater and waste solutions because of its ease of use and economics. The concentration and content of the compounds, the solution matrix, the process efficiency, and the material adsorption capability all influence the choice of adsorbent material.

The behavior of the substance with adsorbent qualities, from a kinetic and thermodynamic perspective, determines the effectiveness of the adsorption process. Because it takes longer for adsorbed molecules to enter the adsorbent particles, a solid material with a high adsorption capacity but a low reaction rate is not a viable choice. On the contrary, a high response rate adsorbent with a limited adsorption capacity is not advantageous because it requires a large quantity of adsorbent, which would result in additional costs. The optimum material for the adsorption process is an adsorbent that possesses both a high adsorption capacity and a rapid response rate.

To recover Ag(I) from aqueous solutions, the adsorptive qualities of SFL, SFC, and SFB materials were explored. Static adsorption experiments were conducted for this reason. The following factors were examined as they affect the adsorption process: temperature, contact time, adsorbent dose, solution pH, and initial Ag(I) concentration. Subsequently, the ideal adsorption conditions were determined to be pH > 4, contact time 60 min, temperature 298 K, and solid–liquid ratio (S–L) of 0.1 g of adsorbent to 25 mL of Ag (I) solution for all three materials. It is possible to determine whether the adsorption processes under investigation are physical in nature, spontaneous, and endotherm, and if they occur in a monolayer on the basis of the data obtained from the kinetic, thermodynamic, and equilibrium investigations conducted. The Langmuir isotherm is the one that best describes the experimental data. The maximum adsorption capacities of the materials, 19.7 mg Ag(I)/g for SFC, 19.3 mg Ag(I)/g for SFB, and 19.9 mg Ag(I)/g for SFL, are comparable. The adsorption mechanism is physical in nature, as evidenced by the activation energies of 1.6 kJ/mol for SFC, 4.15 kJ/mol for SFB, and 1.32 kJ/mol for SFL. The highest Ag(I) concentration used for doping all three materials in the study was 150 mg Ag(I)/L.

The materials doped with silver ions were employed as antimicrobial materials after post-adsorption characterizations showed that Ag(I) was successfully adsorbed on them. In terms of the real application for these materials, the antibacterial and antifungal properties of packaging or storage spaces, especially in terms of industrial uses, are very important. Subsequently, the antimicrobial activity against *Staphylococcus aureus*, *Escherichia coli*, *Pseudomonas aeruginosa*, and *Candida albicans* microorganisms was evaluated by rate of inhibition assessment. The SFC-Ag material showed a percentage of 100% inhibition with respect to the positive control for each microorganism. The fact that the synthetized material has good and very good antifungal effect, it is a desideratum to consider it an efficient antifungal agent for some industrial or medical purposes.

Thus, it was found that silver doping of a composite material consisting of individual elements with varying properties is a promising strategy for obtaining materials with directed properties and broad-spectrum antimicrobial activity for various applications.

## Figures and Tables

**Figure 1 toxics-13-00825-f001:**
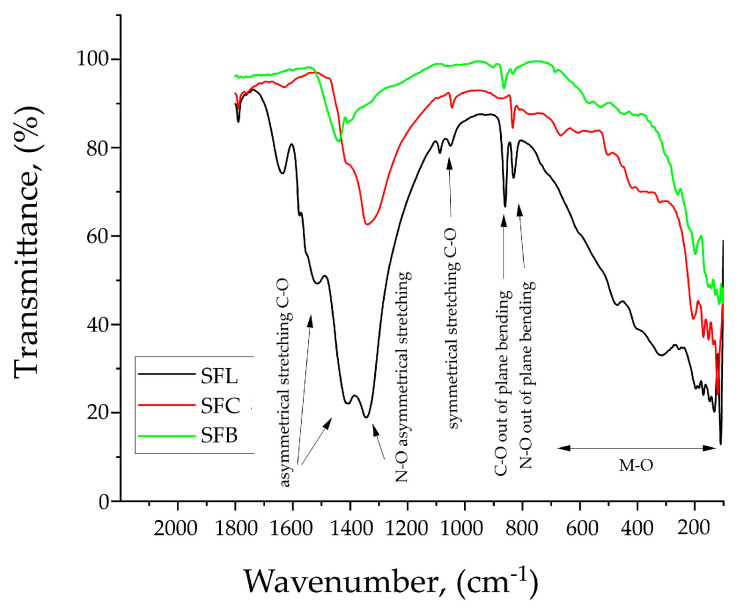
FT-IR spectra of the synthesized materials.

**Figure 2 toxics-13-00825-f002:**
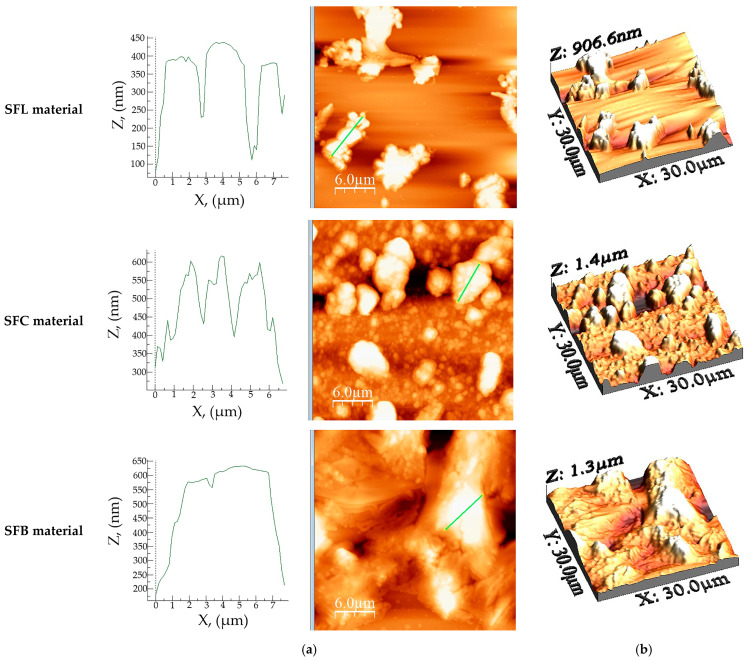
Atomic force microscopy for the synthesized materials. (**a**) Profile on selected area, (**b**) 3D image of the area.

**Figure 3 toxics-13-00825-f003:**
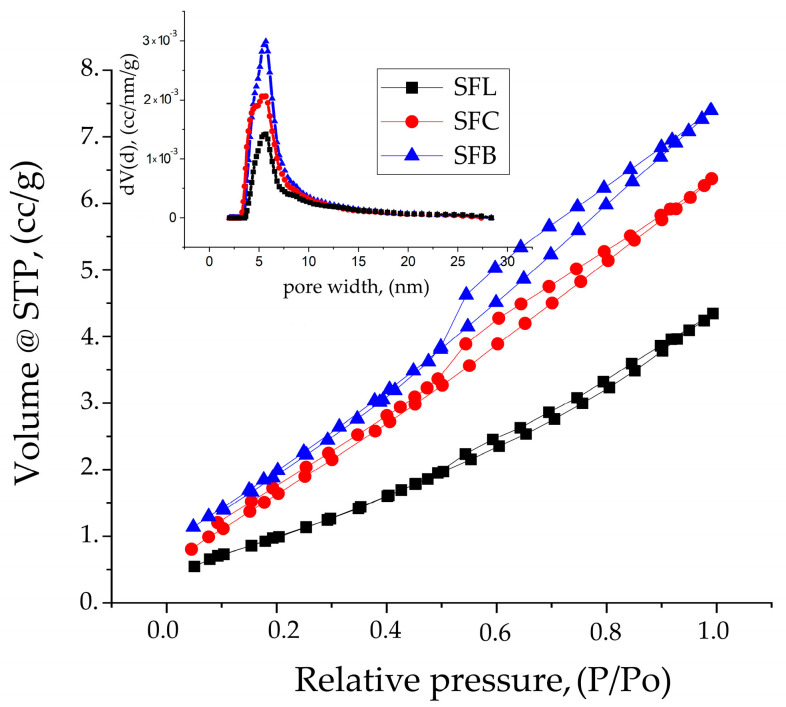
Pore volume and pore width (inset) calculated using the Barrett–Joyner–Halenda (BJH) model from the desorption branch of the isotherms for the SFL, SFC, and SFB samples.

**Figure 4 toxics-13-00825-f004:**
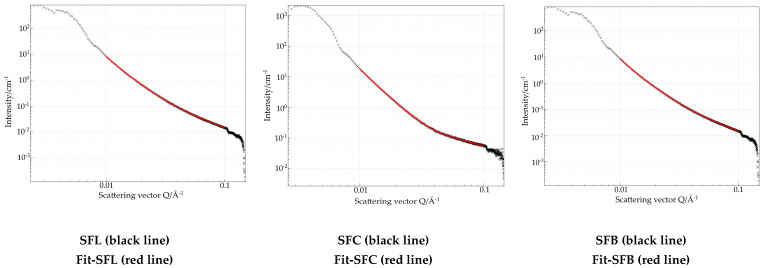
Small-angle X-ray scattering for the synthesized materials.

**Figure 5 toxics-13-00825-f005:**
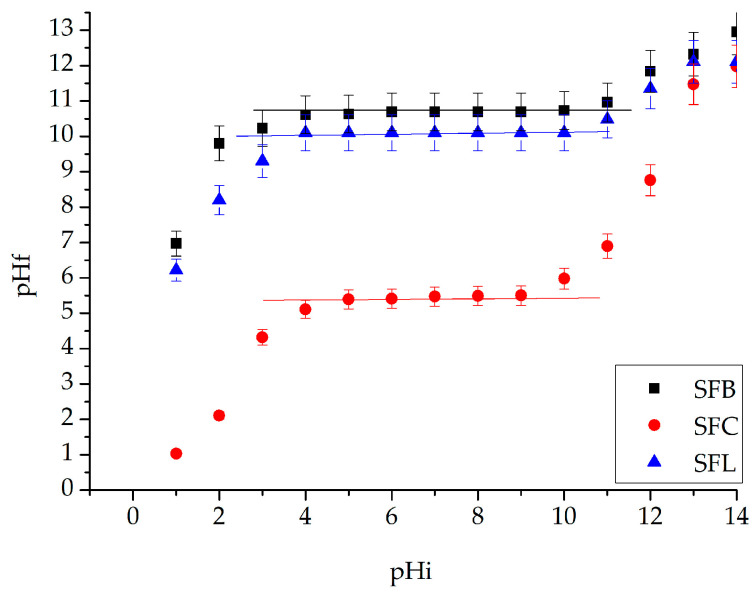
pH_pZc_ for the synthesized materials.

**Figure 6 toxics-13-00825-f006:**
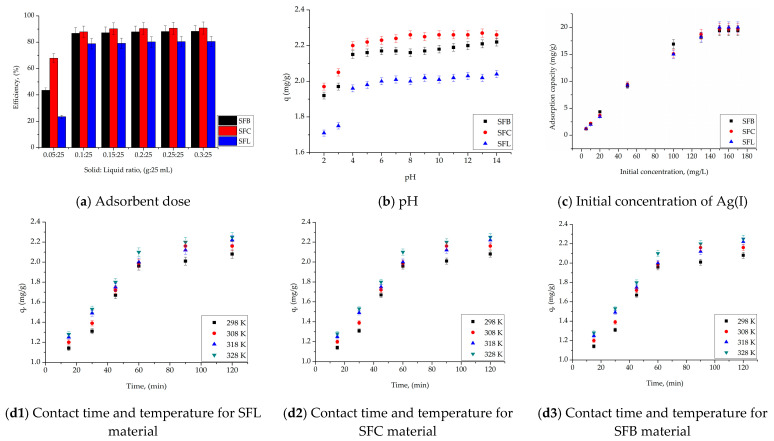
Adsorption parameters’ effect (adsorbent dose, (**a**), pH, (**b**), initial concentration of Ag(I) (**c**), contact time and temperature (**d1**–**d3**)) for materials.

**Figure 7 toxics-13-00825-f007:**
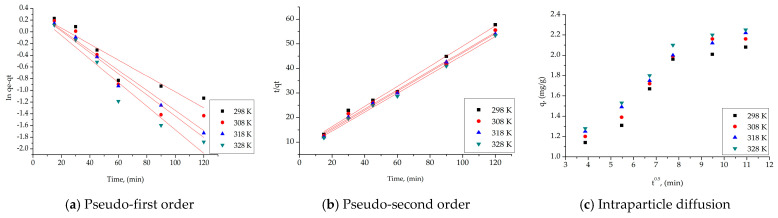
Kinetic studies for the silver adsorption process onto SFL, SFC, and SFB.

**Figure 8 toxics-13-00825-f008:**
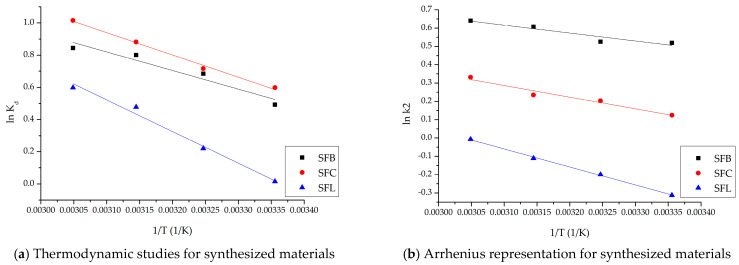
Thermodynamic studies.

**Figure 9 toxics-13-00825-f009:**
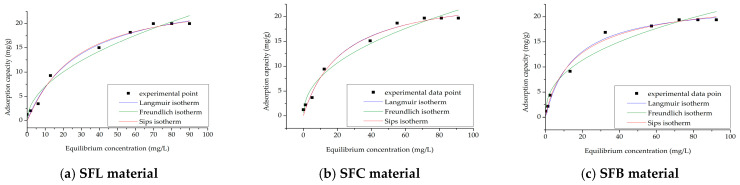
Adsorption isotherms for synthetized materials.

**Figure 10 toxics-13-00825-f010:**
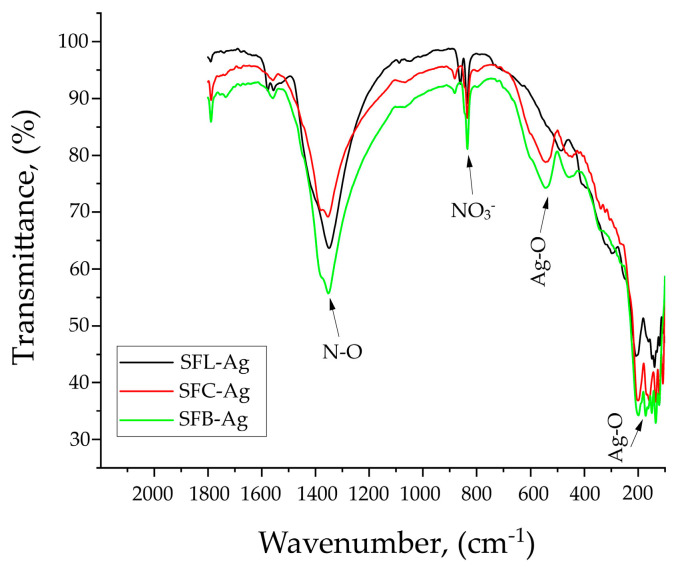
FT-IR spectra after doping the materials with silver.

**Figure 11 toxics-13-00825-f011:**
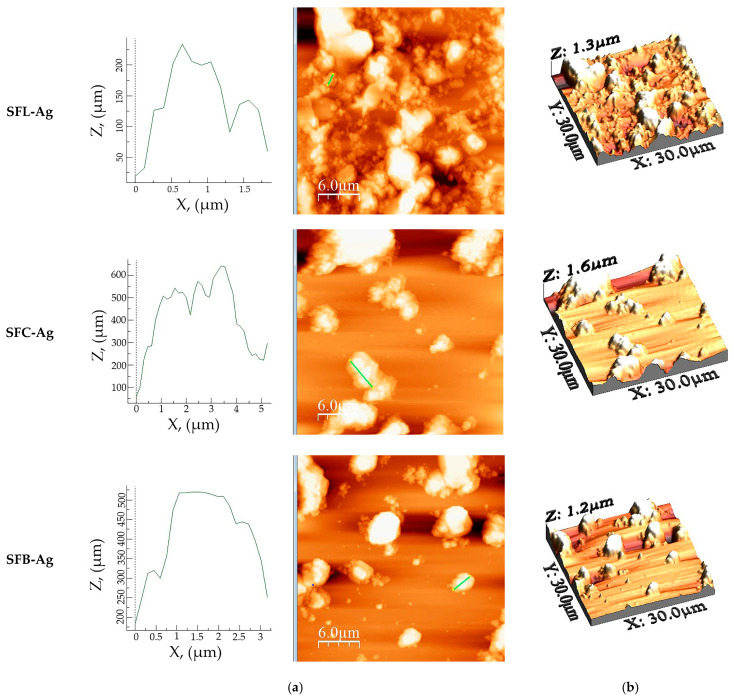
Atomic force microscopy for materials doped with silver. (**a**) Profile on selected area, (**b**) 3D image of the area.

**Figure 12 toxics-13-00825-f012:**
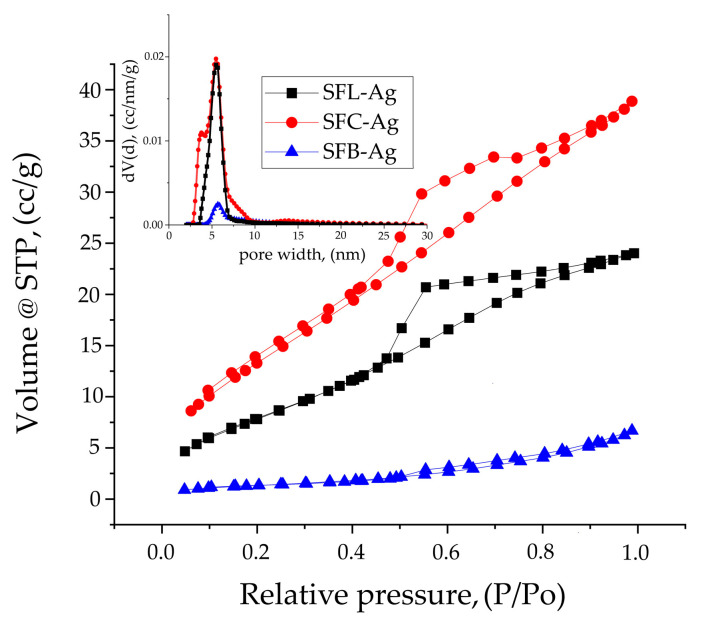
BET for the materials doped with silver.

**Figure 13 toxics-13-00825-f013:**
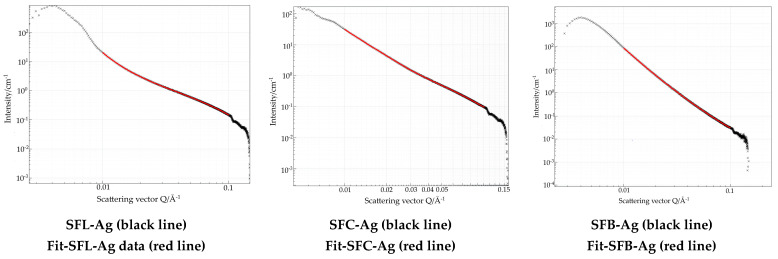
SAXS measurements for the materials doped with silver.

**Table 1 toxics-13-00825-t001:** Comparison of various adsorbents on adsorption of Ag(I).

Adsorbents	Adsorption Capacity, mg/g	References
Biochar	43.9	[28]
Bentonite	55.6	[29]
Immobilized crab shell beads	2.9	[30]
Immobilized coffee ground beads A	36.3	[31]
Vermiculite	69.2	[32]
Zeolite	446.7	[33]
Silica nanosheets functionalized by azole derivatives	139.5	[34]
MgO2-coal	93.57	[35]
MOFs (Fe3O4@UiO−66)	226.88	[36]
Fe3O4@SiO2@CaSiO3	127.84	[37]
Biomass Adsorbent	23.92	[38]
Fe3O4@SiO2@TiO2−IIP	35.475	[39]
Ag-imprinted polymer, Ag-IIP	296	[40]
Poly(o-phenylenediamine) microparticles	533	[41]
SFC	19.7	This paper
SFB	19.3	This paper
SFL	19.9	This paper

**Table 2 toxics-13-00825-t002:** Values obtained from AFM analysis.

Sample Name	Ironed Area, (µm^2^)	*S_a_*, (nm)	*S_q_*, (nm)	*S_p_*, (nm)	*S_v_*, (nm)	*S_y_*, (nm)	*S_ku_*	*S_sk_*
SFL	945	119.8	166	442.6	−464	906.6	3.592	0.762
SFC	1024	221	287	703	−674	1377	3.062	0.438
SFB	939	193.7	251	634	−619	1253	3.074	0.176

**Table 3 toxics-13-00825-t003:** Textural parameters for the materials synthesized.

Sample Name	Surface Area, (m^2^/g)	Pore Width DFT, (nm)	Total Pore Volume, (cc/g)	FHH, D
SFL	4.4	5.682	6.454 × 10^−3^ for pores smaller than 87.8 nm	1.9603
SFC	7.1	5.682	9.874 × 10^−3^ for pores smaller than 220.7 nm	1.9118
SFB	8.2	5.682	1.147 × 10^−2^ for pores smaller than 191.6 nm	1.9382

**Table 4 toxics-13-00825-t004:** Size distributions from scattering data.

Sample Name	Median of the Distribution, (nm)	Surface Equivalent Mean Size, (nm)	Fractal Dimension
SFL	460	15	2.6
SFC	289	9	2.3
SFB	552	28	2.6

**Table 5 toxics-13-00825-t005:** Kinetic parameters.

SFL material					
Pseudo-first order					
Temperature (K)	*q_e,exp_* (mg g^−1^)	*k*_1_ (min^−1^)		*q_e,calc_* (mg g^−1^)	*R* ^2^
298	2.01	0.0136		1.39	0.8838
308	2.16	0.0161		1.42	0.9141
318	2.17	0.0181		1.45	0.9683
328	2.20	0.0201		1.46	0.9451
Pseudo-second order					
Temperature (K)	*q_e,exp_* (mg g^−1^)	*k*_2_ (g mg^−1^∙min^−1^)		*q_e,calc_* (mg g^−1^)	*R* ^2^
298	2.01	0.73		2.44	0.9902
308	2.16	0.81		2.53	0.9918
318	2.17	0.89		2.56	0.9960
328	2.20	0.99		2.60	0.9956
Intraparticle diffusion model (IPD)					
Temperature (K)	*K_diff_* (mg·g^−1^ min^−1/2^)		*C*		*R* ^2^
298	0.053		1.21		0.8531
308	0.076		1.28		0.8754
318	0.089		1.31		0.8890
328	0.096		1.46		0.8632
SFC material					
Pseudo-first order					
Temperature (K)	*q_e,exp_* (mg g^−1^)	*k*_1_ (min^−1^)		*q_e,calc_* (mg g^−1^)	*R* ^2^
298	2.21	0.0194		1.17	0.8376
308	2.24	0.0201		1.10	0.8086
318	2.27	0.0235		1.18	0.8338
328	2.30	0.0251		1.124	0.8287
Pseudo-second order					
Temperature (K)	*q_e,exp_* (mg g^−1^)	*k*_2_ (g mg^−1^∙min^−1^)		*q_e,calc_* (mg g^−1^)	*R* ^2^
298	2.21	1.13		2.55	0.9899
308	2.24	1.22		2.56	0.9915
318	2.27	1.26		2.60	0.9919
328	2.30	1.39		2.59	0.9939
Intraparticle diffusion model (IPD)					
Temperature (K)	*K_diff_* (mg·g^−1^ min^−1/2^)		*C*		*R* ^2^
298	0.083		1.44		0.8726
308	0.086		1.48		0.8430
318	0.089		1.54		0.8915
328	0.092		1.56		0.8751
SFB material					
Pseudo-first order					
Temperature (K)	*q_e,exp_* (mg g^−1^)	*k*_1_ (min^−1^)		*q_e,calc_* (mg g^−1^)	*R* ^2^
298	2.18	0.0128		1.23	0.8049
308	2.24	0.0167		1.24	0.8270
318	2.25	0.0178		1.25	0.8284
328	2.29	0.0205		1.23	0.8381
Pseudo-second order					
Temperature (K)	*q_e,exp_* (mg g^−1^)	*k*_2_ (g mg^−1^∙min^−1^)		*q_e,calc_* (mg g^−1^)	*R* ^2^
298	2.18	1.68		2.34	0.9969
308	2.24	1.69		2.42	0.9961
318	2.25	1.83		2.44	0.9969
328	2.29	1.89		2.47	0.9968
Intraparticle diffusion model (IPD)					
Temperature (K)	*K_diff_* (mg·g^−1^ min^−1/2^)		*C*		*R* ^2^
298	0.080		1.37		0.8229
308	0.083		1.39		0.8310
318	0.087		1.43		0.8305
328	0.088		1.47		0.8351

**Table 6 toxics-13-00825-t006:** Thermodynamic parameters specific to the adsorption process.

Δ*H*° (kJ/mol)	Δ*S*° (J/mol∙K)	Δ*G*° (kJ/mol)					*R* ^2^
		298 K	308 K	318 K	328 K		
SFL material							
16.33	54.98	−16.3	−16.9	−17.4		−18.0	0.9868
SFC material							
11.49	45.46	−12.9	−13.3	−13.8		−14.2	0.9940
SFB material							
9.59	36.57	−10.8	−11.2	−11.6		−11.9	0.9802

**Table 7 toxics-13-00825-t007:** Activation energy.

Materials	*E_a_* (kJ/mol)	*R* ^2^
SFL	1.32	0.9984
SFC	1.60	0.9642
SFB	4.15	0.9904

**Table 8 toxics-13-00825-t008:** Parameters of the isotherm model.

Materials	Langmuir Isotherm			
	*q_m,exp_* (mg/g)	*K_L_* (L/mg)	*q_L_* (mg/g)	*R* ^2^
SFL	19.9	0.0336	24.4	0.9882
SFC	19.7	0.0435	25.4	0.9878
SFB	19.3	0.0648	23.1	0.9854
	Freundlich isotherm			
	*K_F_* (mg/g)	1/*n_F_*		*R* ^2^
SFL	2.13	0.51		0.9715
SFC	2.67	0.46		0.9722
SFB	3.38	0.40		0.9613
	Sips isotherm			
	*K_S_*	*q_S_* (mg/g)	1/*n_S_*	*R* ^2^
SFL	0.03	25.7	0.07	0.9865
SFC	0.04	25.7	0.02	0.9858
SFB	0.07	25.0	0.03	0.9843

**Table 9 toxics-13-00825-t009:** Extracted data parameters for the morphology of materials doped with silver.

Sample Name	Ironed Area (µm^2^)	*S_a_* (nm)	*S_q_* (nm)	*S_p_* (nm)	*S_v_* (nm)	*S_y_* (nm)	*S_ku_*	*S_sk_*
SFL-Ag	1010	222	279	681	−667	1348	2.821	0.390
SFC-Ag	973	188	275	710	−846	1556	4.061	0.316
SFB-Ag	960	147	216	604	−562	1166	3.936	0.724

**Table 10 toxics-13-00825-t010:** Textural parameter for the doped materials with silver.

Material	Surface Area, (m^2^/g)	DFT Pore Width, (nm)	Total Pore Volume, (cc/g)	FHH, D
SFL-Ag	30.9	5.483	3.726 × 10^−2^ cc/g for pores smaller than 249.6 nm	2.1057
SFC-Ag	53.1	5.483	6.028 × 10^−2^ cc/g for pores smaller than 160.4 nm	2.1137
SFB-Ag	4.9	5.682	1.037 × 10^−2^ cc/g for pores smaller than 161.6 nm	1.7725

**Table 11 toxics-13-00825-t011:** Extract size distributions from scattering data for materials doped with silver.

Material	Median of the Distribution (nm)	Surface Equivalent Mean Size (nm)	Fractal Dimension
SFL-Ag	436	7	3.8
SFC-Ag	258	11	3.3
SFB-Ag	593	66	2.2

**Table 12 toxics-13-00825-t012:** Antimicrobial activity of iron–carbonate composites (Li, Cu, Bi) doped with Ag (I).

Material	Ag mg/L	Inhibition Rate (%)				OBS.
		*S. aureus* *ATCC 25923*	*P. aeruginosa* *ATCC 27853*	*E. coli* *ATCC 25922*	*C. albicans* *ATCC 10231*	
SFL-Ag	1	50	50	20	100	Slightly better bactericidal effect on Gram-negative bacteria Very good antifungal activity
SFL-Ag	10	20	50	100	100	
SFL-Ag	50	20	100	100	100	
SFL-Ag	150	20	100	100	100	
SFC-Ag	1	100	100	100	100	Very good antibacterial and antifungal activity
SFC-Ag	10	100	100	100	100	
SFC-Ag	50	100	100	100	100	
SFC-Ag	150	100	100	100	100	
SFB-Ag	1	20	20	20	75	Good antifungal activity Lower antibacterial activity regardless of the type of bacteria
SFB-Ag	10	20	20	20	75	
SFB-Ag	50	20	20	30	90	
SFB-Ag	150	20	60	30	100	

## Data Availability

Data are contained within the article.

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
