# Peer review of "Iron–Carbonate (Bi, Cu, Li) Composites with Antimicrobial Activity After Silver(I) Ion Adsorption"

_toxics, 2025, doi:10.3390/toxics13100825_

Round 1
Reviewer 1 Report
Comments and Suggestions for Authors
The Authors present a relevant study of the antimicrobial effect of the Ag-adsorbed iron nitrate in combination with bis-26muth, copper or lithium carbonates. Paper is well-organized with the well designed pictures and tables. It could be published after the elucidation of some critical points.
My first question refers to the study's concept. It is obvious and clear that the Ag is suggested as the antimicrobial agent. But I would like to suggest the Authors to extend the paper with the description why did the Authors use the iron nitrate modified by carbonates. What is the role of these compounds?
Please, find below other comments.
Line 159. What is the final form of the samples (powder or something else)? did the Authors study the morphology and the grain size of the samples?
Line 491. A typo occured
Fig.1. The region of 600-200 cm-1 should be described in more details. In my opinion, the Authors could obtain an additional information of the metal bondings. Moreover, not only metal-oxygen bonds represent the absorbance in this region.
Section Atomic Force Microscopy.
I suggest the Auhors to use subscript notation for indicies in the Section of Atomic forse microscope study. Also, in the Material Characterization section it is noted that the samples have been heat treated at two different temperatures, but AFM results represent only one image for each sample.
Besides it is unclear how AFM data correspond to he SAXS measurements. According to SAXS the Median of the
distribution equals 289-552 nm, but AFM images show the objects with the size larger than 1 micron. Could the Authors clarify this point?
Author Response
Thank you for your kindness in reading our article and for your valuable recommendations.
We have tried to respond as clearly as possible to your recommendations, so that the quality of our article is appropriate for publication in such a prestigious journal.
1.My first question refers to the study's concept. It is obvious and clear that the Ag is suggested as the antimicrobial agent. But I would like to suggest the Authors to extend the paper with the description why did the Authors use the iron nitrate modified by carbonates. What is the role of these compounds?
Answer: Consequently, the study discusses a composite material, which is a combination of multiple components, rather than a single, specific chemical composition. This is significant because such a material would likely leverage the unique properties of each ingredient: carbonates for their adsorptive capabilities, bismuth and copper for their antimicrobial effects and potential roles in silver binding, and iron for possible magnetic separation or redox reactions. Its application in fields like bioremediation, biosorption, or the regulation of microbiological growth while recovering silver is suggested by its microbiological uses.
So, the study can be considered a closed-loop, sustainable study.
2. Line 159. What is the final form of the samples (powder or something else)? did the Authors study the morphology and the grain size of the samples?
Answer: The final form of the samples is powder. It was specified in line 168.
Yes, the synthesized materials were characterized physicochemically, using infrared spectrometry, atomic force microscopy, and porosity and surface area were determined. The results regarding the size, shape and morpho-structural properties are presented in the manuscript, in the Results and Discussion section.
3. Line 491. A typo occurred.
Answer: Corrected in the manuscript.
4. Fig.1. The region of 600-200 cm-1 should be described in more detail. In my opinion, the Authors could obtain additional information on the metal bondings. Moreover, not only on metal-oxygen bonds represent the absorbance in this region.
Answer: The FT-IR spectrum in the range of 200-600 cm⁻¹ is particularly relevant for identifying bonds between a metal (M) and a non-metal atom, especially with oxygen (O). This frequency range, also called the infrared fingerprint region, contains stretching and bending vibrations of the M-O bonds.
More details were added in lines 412-416.
5. I suggest the Auhors to use subscript notation for indicies in the Section of Atomic forse microscope study.
Answer: As per request, in the manuscript text.
6. Also, in the Material Characterization section it is noted that the samples have been heat treated at two different temperatures, but AFM results represent only one image for each sample.
Answer: It was a mistake. The text was corrected.
7. Besides it is unclear how AFM data correspond to he SAXS measurements. According to SAXS the Median of the distribution equals 289-552 nm, but AFM images show the objects with the size larger than 1 micron. Could the Authors clarify this point?
Answer: In the manuscript text, the explanation was related, from line 487.
This is also related below:
AFM provides images of the sample's topography in real space, while SAXS offers statistical, average information about size, shape, and aggregation states from a larger sample volume in reciprocal space. Since AFM obtained data refers only to the scanned area (with the maximum possibility of scanning a 50x50 μm area), some differences in the case of SAXS data can be expected. The correlation lies in how they provide different but consistent perspectives, with AFM's real-space visualizations confirming SAXS data and “aiding model interpretation, thus increasing the reliability and providing a more complete understanding of the sample's overall and local structure. The discrepancy between the SAXS and AFM data is common as suggested by Christopher K. et all.
Although the median data obtained from the SAXS and observed dimensions in AFM images are not similar for the same reason explained above, relating to the different method of measurement whereas AFM is direct while SAXS generates data in reciprocal space, requiring modeling, Fourier transformations, or fitting to obtain real-space structural information, the fractal dimension data obtained from SAXS and surface kurtosis data confirm the prevalent rough surface with high peaks and low valleys.
Reviewer 2 Report
Comments and Suggestions for Authors
The authors have presented an article about iron-carbonate (bi, cu, li) composites with antimicrobial activity after silver(i) ion adsorption. However, some issues need addressing before the manuscript is suitable for publication.
- I ask the authors to graphically present the structures of the obtained composites, taking into account individual chemical bonds.
- I ask the authors to provide a graphical representation of the adsorption mechanism in the experimental systems studied.
- I would suggest additionally examining the thermal properties of the obtained composites.
- I would also suggest performing adsorption tests on individual composite components.
- Please enrich your applications with numerical data.
- Please standardize the entire text with the correct notation of the chemical formulas, units, spaces, etc.

Author Response
Thank you for reading our article and for your valuable recommendations.
We have tried to respond as clearly as possible to your recommendations, so that the quality of our article is appropriate for publication in such a prestigious journal.
1.I ask the authors to graphically present the structures of the obtained composites, taking into account individual chemical bonds.
Answer:
To establish a certain mechanism, more intermediate characterizations would be necessary, at various stages of the synthesis.
The process involves several steps, culminating in controlled precipitation, which leads to the formation of metal-oxygen (M-O) bonds.
Dissolution of precursors: Initially, copper carbonates (CuCO3), bismuth carbonates (Bi2(CO3)2·(OH)2) and lithium carbonates (Li2CO3) are dissolved in a mixture of water and methanol, acidified with nitric acid (HNO3). The nitric acid reacts with carbonates to form soluble nitrates.
- CuCO3+2HNO3→Cu(NO3)2+H2O+CO2
- Bi2(CO3)·2(OH)2+6HNO3→2Bi(NO3)3+4H2O+2CO2
- Li2CO3+2HNO3→2LiNO3+H2O+CO2
Iron nitrate (Fe(NO3)3) is added to the solution. At this point, the solution contains free metal ions: Cu2+, Bi3+, Li+, Fe3+, and nitrate ions (NO3-).
The added sodium hydroxide solution quickly raises the pH of the mixture. As the pH increases, the metal ions in the solution combine with hydroxide ions (OH−) to form metal hydroxides, which are usually insoluble.
- yOH−→M(OH)x↓ (where M is Cu, Bi, Li, Fe)
The precipitated product, which is a mixture of hydroxides, is dried at 100°C. This temperature is sufficient to eliminate free water. However, in many cases, at these temperatures, partial M-O-M bonds can form, especially in the presence of several metal ions that can form a mixed compound.
The main bond that is formed is the covalent or ionic one between the metal and oxygen atoms. These can be found in structures such as mixed metal oxides (e.g. CuFeO2, LiFeO2, BiFeO3), or as a mixture of individual oxides.
Given that the precipitation occurred by adding NaOH, it is very likely that metal hydroxides are present in the dried product. Therefore, M-OH bonds can exist only if the drying process at 100°C was not sufficient to eliminate all the water.
Hydrogen bonds can be formed between water molecules, hydroxyl groups, and possibly between incompletely reacted nitrate or carbonate residues.
In conclusion, the final compounds are a complex mixture of phases, predominantly metal oxides and hydroxides.
The main chemical bonds are M-O-M (within the crystalline phase) and M-OH, which is consistent with FT-IR analysis, which indicated the presence of M-O and, most likely, C-O bond vibrations.
2.I ask the authors to provide a graphical representation of the adsorption mechanism in the experimental systems studied.
Answer:
The mechanism was proposed in the abstract graph of the paper. I have attached this graph.
3. I would suggest additionally examining the thermal properties of the obtained composites.
Answer:
Thanks for the suggestion. Studies of the thermal properties of materials are very important. These have been carried out, but given the size of the manuscript, we have decided not to publish these results.
4. I would also suggest performing adsorption tests on individual composite components.
Answer:
Adsorption studies were performed for each synthesized material ((FeNO3@Li2CO3 (SFL), FeNO3@CuCO3 (SFC), and FeNO3@(BiO)2CO3 (SFB)) in part. The behavior is similar. As follows: The maximum adsorption capacities of all the materials were: 19.9 mg Ag(I)/g for SFL, 19.7 mg Ag(I)/g for SFC, and 19.3 mg Ag(I)/g for SFB for the same initial concentration, Ci=150 mg Ag(I)/L (data from paper starting in line 575).
5. Please enrich your applications with numerical data.
Answer:
Thank you for the recommendation. The data were added to the manuscript in the conclussion section.
6. Please standardize the entire text with the correct notation of the chemical formulas, units, spaces, etc.
Answer:
Thank you for the recommendation.
The manuscript was checked, and we tried to solve all the improper notation, units, and spaces.

Round 2
Reviewer 1 Report
Comments and Suggestions for Authors
I thank the Authors for the provided comments and the revised version.
Neverthelessm I do not agree with the explanation of the discrepancy between AFM and SAXS data.
Author Response
We believe there was a misunderstanding! Our goal was not to compare the values obtained from AFM analysis with those from SAXS analysis, but to gather information about the surface's roughness or smoothness.
The results of the three different measurements, obtained by three distinct experimental methods, demonstrated a similar behavior. Specifically: 1. Values for the roughness parameter Sku were obtained using atomic force microscopy (AFM). 2. Values for the fractal dimension D were obtained using two different methods: one based on adsorption (FHH) and one based on X-ray scattering (SAXS). 3. All these data sets (Sku from AFM, D from FHH and D from SAXS) were compared. The conclusion is that although they are different parameters measured by various methods, they all indicate the same trend. For example, if the samples were treated in a certain way, both Sku and D (regardless of the measurement method) increased or decreased consistently, indicating a similar modification of the surface. This concordance of the results is solid evidence of the validity of the experiments and the conclusion drawn, as it demonstrates that the observed phenomenon is real and not an artifact of a single measurement method.

Reviewer 2 Report
Comments and Suggestions for Authors
The authors responded honestly to the comments made during the review, took into account the reviewer's suggestions and made appropriate corrections to the manuscript.
Author Response
Thank you for your motivating feedback for us as authors.